# Understand Before You Generate: Self-Guided Training for Autoregressive Image Generation

**Xiaoyu Yue**[1,2]    **Zidong Wang**[3]    **Yuqing Wang**[4]    **Wenlong Zhang**[1]
**Xihui Liu**[4]    **Wanli Ouyang**[1,3]    **Lei Bai**[1]    **Luping Zhou**[2]
[1]Shanghai AI Laboratory    [2]University of Sydney
[3]Chinese University of Hong Kong    [4]University of Hong Kong

## Abstract

Recent studies have demonstrated the importance of high-quality visual representations in image generation and have highlighted the limitations of generative models in image understanding. As a generative paradigm originally designed for natural language, autoregressive models face similar challenges. In this work, we present the first systematic investigation into the mechanisms of applying the next-token prediction paradigm to the visual domain. We identify three key properties that hinder the learning of high-level visual semantics: local and conditional dependence, inter-step semantic inconsistency, and spatial invariance deficiency. We show that these issues can be effectively addressed by introducing self-supervised objectives during training, leading to a novel training framework, **S**elf-guided **T**raining for **A**uto**R**egressive models (ST-AR). Without relying on pre-trained representation models, ST-AR significantly enhances the image understanding ability of autoregressive models and leads to improved generation quality. Specifically, ST-AR brings approximately $42\%$ FID improvement for LlamaGen-L and $49\%$ FID improvement for LlamaGen-XL, while maintaining the same sampling strategy[1].

## 1 Introduction

The field of image generation has witnessed remarkable progress through various approaches, including diffusion models [27, 43, 32, 52, 18, 37, 34, 41, 33, 38, 20], Generative Adversarial Networks (GANs) [21, 22, 42], and autoregressive models (AR) [19, 44]. Among these, autoregressive models, originally developed for natural language processing (NLP), have demonstrated exceptional generative capabilities as the foundational paradigm for large language models (LLMs) such as GPT [2] and Llama [47, 48]. When adapted to image generation, autoregressive models achieve performance comparable to modality-specific methods, indicating their potential as a unified generative framework across diverse data modalities [5, 8, 16, 45, 49].

Recent studies have highlighted the importance of image understanding in enhancing generation performance. For instance, REPA [54] enhances the generative capabilities of diffusion models [34, 51] by distilling self-supervised representations into their intermediate layers. Similarly, ImageFolder [31] introduces semantic regularization to the quantizer of the tokenizer to inject semantic constraints. These methods rely on pre-trained representation models to provide additional semantic information, as denoising and compressing may not be appropriate tasks for learning semantically meaningful image representations [54]. In contrast, the next-token prediction paradigm used by autoregressive models has proven to be an effective pre-training approach for capturing contextual information in natural language processing [39, 28, 40]. However, when adapted to vision, due to the inherent differences between image and text modalities, next-token prediction also faces challenges in learning high-level visual representations.

---

[1]Project Page: `https://github.com/yuexy/ST-AR`

39th Conference on Neural Information Processing Systems (NeurIPS 2025).

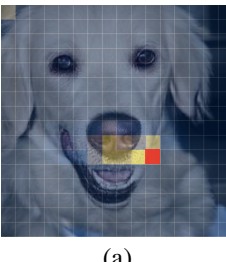 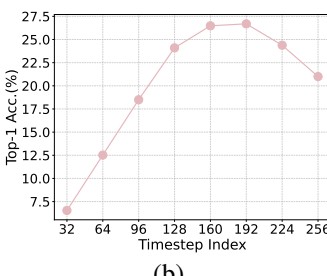 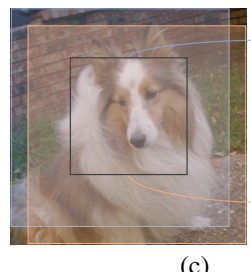 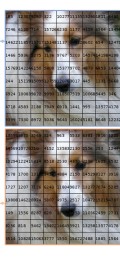

|  (a) | (b) | (c) |

Figure 1: **Illustration of three properties of LlamaGen-B model.** (a) Attention map from the last layer, highlighting the current token in red and tokens with larger attention weights in yellow. (b) Linear probing results on features from the $6\text{-}th$ layer at 8 uniformly selected steps. (c) Visual token indices from two slightly different views of the same image.

In this work, we aim to *enhance the learning of high-level visual representations in autoregressive models to improve the generative capability.* Employing the popular autoregressive model Llama-Gen [44], we first conduct an in-depth investigation into the intrinsic mechanisms of autoregressive image generation and identify three key properties that impact visual understanding:

*(1) Local and conditional dependence.* Autoregressive models predominantly depend on local and conditional information. Our analysis of attention maps, as shown in Figure 1 (a), reveals a strong dependence on the initial step (conditioning token) and spatially adjacent steps, highlighting that the model primarily utilizes conditional and local information for its predictions.

*(2) Inter-step semantic inconsistency.* Figure 1 (b) demonstrates inconsistent semantic information across different timesteps, as evidenced by the top-1 linear probing accuracy. Specifically, while accuracy increases in early timesteps with more visible image tokens, its subsequent decline reveals that autoregressive models fail to maintain previously learned semantic information, thereby limiting the global modeling capability.

*(3) Spatial invariance deficiency.* Autoregressive image generation models typically employ visual tokenizers, such as VQ-GAN [19, 29] to quantize images into discrete tokens. However, slight perturbations in image space can result in completely different tokens, as shown in Figure 1 (c). This ambiguity of objects significantly increases the difficulty for autoregressive models in encoding visual signals.

These three problems create bottlenecks for autoregressive models in learning high-quality image representations, mirroring the challenge faced by diffusion models as revealed by REPA [54]. To this end, we propose **ST-AR**, short for **S**elf-guided **T**raining for **A**uto**R**egressive models, a novel training paradigm that leverages techniques well-explored in self-supervised learning to enhance the modeling of visual signals. Specifically, for property 1, inspired by masked image modeling [53, 25] that forces the network to attend to larger regions of the image [36, 55], we randomly mask a portion of the tokens in the attention map of the transformer layers. Meanwhile, for properties 2 & 3, we employ contrastive learning [15] to ensure the consistency of feature vectors from different time steps and views, referred to as inter-step contrastive loss and inter-view contrastive loss, respectively. The resulting training paradigm, ST-AR, incorporates a MIM loss and two contrastive losses in addition to the token prediction loss, forming an iBOT-style [56] framework. ST-AR is utilized only during training, and the trained models retain the autoregressive sampling strategy, thus preserving their potential for unification with other modalities.

By integrating visual self-supervised paradigms into next-token prediction, ST-AR eliminates the need for pre-trained representation learning models to provide additional knowledge, achieving stronger image understanding solely through self-guided training. Specifically, ST-AR significantly improves the linear probing top-1 accuracy of LlamaGen-B from $21.00\%$ to $55.23\%$ and demonstrates semantically meaningful attention maps. Furthermore, the enhancement in image understanding facilitates image generation. On class-conditional ImageNet, ST-AR boosts LlamaGen-B by 7.82 FID score. Notably, LlamaGen-XL trained with ST-AR for just 50 epochs achieves approximately a $49\%$ improvement in FID over the baseline, and is even comparable to LlamaGen-3B trained for 300 epochs, despite the latter having about $4\times$ more parameters.

Our contributions can be summarized as follows:

- **Conceptually**, we conduct an in-depth investigation into the mechanisms of autoregressive image generation, identifying three key properties that hinder visual representation learning.

- **Technically**, we propose a novel training paradigm, ST-AR, which enhances image understanding by integrating self-supervised training techniques into the next-token prediction paradigm.

- **Experimentally**, we conduct comprehensive experiments to validate the design of each component of ST-AR, demonstrating its effectiveness in both image understanding and generation.

## 2  Related Work

**Autoregressive Image Generation.**   The autoregressive (AR) generation paradigm has established itself as a leading approach in language modeling [40, 1–4] due to its simplicity, scalability, and zero-shot generalization capabilities. When extended to image generation, AR methods can be categorized into three types according to the sampling strategies. Causal AR methods, such as VQ-GAN [19] and LlamaGen [44], directly adapt AR architectures for image synthesis, utilizing the traditional raster-order next-token prediction paradigm as language models. Masked AR methods, like MaskGiT [11] and MAR [30], employ bi-directional attention within an encoder-decoder framework, supporting iterative generation with flexible orders. Parallelized AR methods introduce vision-specific designs to enhance visual signal modeling capability. VAR [46] proposes next-scale prediction that progressively generates tokens at increasing resolutions. PAR [50] and NPP [35] propose token grouping strategies to generate image tokens in parallel. Although masked and parallelized AR methods enhance the modeling of bidirectional image contexts, they require adjustments to sampling strategies. Our ST-AR focuses on improving the modeling of visual modalities within AR models without altering the sampling strategy, thereby enhancing image generation performance while preserving compatibility with language models.

**Self-Supervised Learning.**   In the field of visual self-supervised learning, methods can be broadly categorized into two types: contrastive learning and masked image modeling. The first to emerge was contrastive learning, exemplified by methods such as SimCLR[12], BYOL[23], MoCo[14, 24, 15], SwAV[9], and DINO[10]. These approaches typically employ image augmentation techniques to construct sets of positive samples and optionally use augmented views from other images as negative samples. They learn semantic information by aligning the representations of positive samples. Masked Image Modeling (MIM) [6, 53, 25] adapts the concept of Masked Language Modeling from NLP, training networks to reconstruct randomly masked portions of image content, thereby learning visual context. Some studies have shown that MIM primarily learns low-level pixel correlations and can adjust the effective receptive field size of the network by modifying the mask ratio. Our ST-AR leverages the strengths of both contrastive learning and MIM, using random masking on attention maps to increase the attention distance of autoregressive models, as well as employing MoCo-like contrastive losses to align representations across different time steps and different views.

## 3  Method

### 3.1  Preliminaries

We provide a brief review of visual autoregressive models operating in discrete space. Given an input image $I$, a quantized autoencoder is employed to convert $I$ to a sequence of discrete tokens: $\mathbf{x} = q(I)$, where $\mathbf{x} = [x_1, x_2, ..., x_T]$ is the output token sequence, and $q(\cdot)$ denote the encoder and quantizer of the quantized autoencoder.

The autoregressive model is trained to maximize the joint conditional probability of predicting the token $x_t$ at the current step $t$, based on the conditional vector $c$ and the preceding tokens $[x_1, x_2, ..., x_{t-1}]$. The condition $c$ can be a class label or a text vector. The training objective can be formalized as:

$$\max_{\theta} p_{\theta}(\mathbf{x}) = \prod_{t=1}^{T} p_{\theta}(x_t|c, x_1, x_2, ..., x_{t-1}), \tag{1}$$

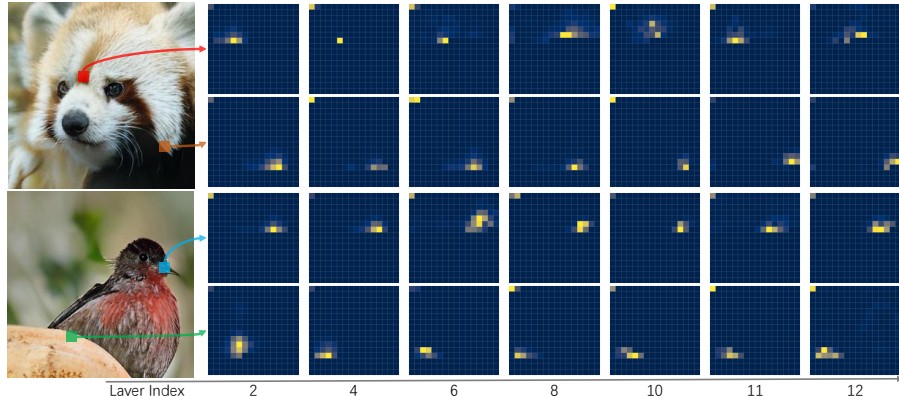

Layer Index    2      4      6      8      10      11      12

Figure 2: **Attention maps of LlamaGen-B across layers and steps.** These attention maps consistently show that conditional and spatially adjacent tokens receive the highest attention weights, while other tokens have significantly lower weights.

where $p_\theta$ is the autoregressive model parameterized by $\theta$. And the token prediction loss is:

$$\mathcal{L}_{AR} = -\frac{1}{T} \sum_{t=1}^{T} \log p_\theta(x_t | c, x_{<t}). \tag{2}$$

After training, $p_\theta$ can iteratively generate new sequences. This process known as the next-token prediction, has been proven effective in text modeling.

## 3.2 Observations

We conduct an in-depth investigation into the intrinsic mechanisms of autoregressive models in image generation, evaluating visual understanding capabilities through two aspects: attention maps and linear probing. For the class-conditional LlamaGen-B model trained on ImageNet [17], we first analyze the behavior of the transformer module by visualizing attention maps. Attention maps reveal what the model relies on for predictions and whether it can capture image context. Then, we evaluate the quality of learned representations by comparing linear probing results across intermediate layers at different time steps. Specifically, we closely followed the training protocol in MAE [25] and set the input class embedding to the null unconditional embedding for classifier-free guidance to prevent knowledge leakage. We uniformly select 8 out of 256 steps and feed the features from the sixth layer at the corresponding steps into trainable linear layers. Our observations are as follows:

*Obs.* 1. **Autoregressive models primarily rely on local and conditional information.** In Figure 2, we present attention maps across various depths and positions, all exhibiting a consistent pattern: the highlighted areas predominantly include spatially adjacent tokens and the first token. As indicated in Eq. 1, the input at the initial step is the conditional token, which significantly influences subsequent sampling, thereby holding considerable importance in the attention maps. Tokens surrounding the current token also receive elevated attention weights due to the inherent locality of images. Despite all preceding tokens being visible during training, the spatial proximity of tokens dictates that the most informative tokens for predicting the current token are typically those nearby. Excessive reliance on local information can impede the generation quality, as minor errors in adjacent tokens may be accumulated for subsequent steps.

*Obs.* 2. **Causal Attention Challenges Bi-directional Image Context Modeling.** The application of causal attention to images presents two critical challenges: *semantic inconsistency across different steps* and *limited global modeling capability*. The inherent sequential nature of causal attention, which restricts each step to accessing only previously generated content, fundamentally limits the model's capacity to capture comprehensive global information. As illustrated in Figure 1 (b), the linear probing accuracy at the initial steps is extremely low, indicating that AR models struggle to establish the correct semantic context in the early steps. Furthermore, the observed deterioration in linear probing performance beyond the $192\text{-}th$ step indicates a progressive semantic misalignment in the learned representations as generation proceeds. This phenomenon underscores a critical limitation in the model's ability to maintain and leverage global contextual information effectively throughout

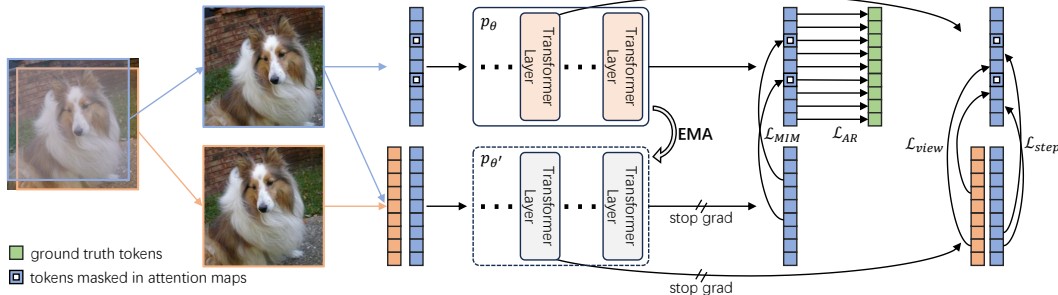

Figure 3: **Overview of Self-Guided Training Pipeline.** We incorporate masked image modeling ($\mathcal{L}_{\text{MIM}}$) to expand the effective field of visual autoregressive models. Additionally, we introduce inter-step contrastive learning ($\mathcal{L}_{step}$) to ensure global consistency, as well as inter-view contrastive learning ($\mathcal{L}_{view}$) for consistency in visual representations.

the generation process. Such constraints pose significant challenges for achieving coherent and semantically consistent image generation.

*Obs.* 3. **Visual tokens lack invariance.** Autoregressive models utilize a visual tokenizer like VQ-GAN to transform continuous image signals into discrete tokens. However, visual tokenizers are primarily trained for image compression and reconstruction, lacking invariance constraints. Consequently, when transformations are applied to an image of a given object, the tokenizer may produce entirely different visual tokens, as demonstrated in Figure 1 (c). This variability in visual signals can confuse the model, resulting in redundant learning of identical semantic concepts.

### 3.3 Self-Guided Training for Autoregressive Models

Building upon these observations, we introduce **S**elf-guided **T**raining for **A**uto**R**egressive models (ST-AR) to enhance the visual understanding capabilities of autoregressive models. ST-AR provides targeted solutions for the aforementioned challenges within a unified training paradigm.

**Overview.** The overall pipeline of ST-AR is illustrated in Figure 3. ST-AR borrows ideas from self-supervised representation learning, employing masked learning to expand attention regions while utilizing contrastive learning to ensure feature alignment across both steps and views. A non-trainable teacher network[14, 24, 10] is employed to provide additional training objectives. It shares the same architecture as the autoregressive model (student model), and weights $\theta'$ are updated through the Exponential Moving Average (EMA) of the student model parameters $\theta$. ST-AR integrates a reconstruction loss and two contrastive losses into the training of autoregressive models, eliminating dependence on pretrained representation models. We refer to it as "Self-Guided Training".

#### 3.3.1 Masked Learning for Longer Contexts

As revealed in [36, 55], masked image modeling (MIM) can expand the effective receptive field of image encoding models. This insight motivates our approach to leverage MIM for addressing the challenge of AR models outlined in *Obs.* 1, *i.e.*, the excessive dependence on local information. However, traditional MIM methods, which substitute input image tokens with a special mask token, is unsuitable for autoregressive models. This is because autoregressive models, unlike autoencoders, necessitate the use of image tokens from the preceding step for next-token prediction. To overcome this, ST-AR utilizes random masking directly on the attention maps within transformer blocks, rather than on the input tokens. A sequence mask $M$ is applied to the attention map, assigning negative infinity (*-inf*) to a ratio $r$ of the total tokens (masked tokens), while normal tokens are assigned as zero. Formally:

$$\text{Attn}(Q_i, K_i, V_i) = \text{Softmax}\left(\frac{Q_i K_i^T}{\sqrt{d_k}} + M\right) V_i, \tag{3}$$

where $Q_i$, $K_i$, and $V_i$ are the query, key, and value matrices for the $i$-th head.

As the masking operation on attention may lead to information loss for next-token prediction, we employ a teacher model to extract features and align the student model accordingly. Specifically, for given input tokens, we solely mask the attention maps of the student model and align the final hidden states of the student network to the teacher network. Given token length $T$, the MIM loss can be

formalized as:

$$\mathcal{L}_{\text{MIM}} = \frac{1}{T} \sum_{t=1}^{T} \mathcal{D}(h_t, \hat{h}_t), \tag{4}$$

where $\mathcal{D}(\cdot, \cdot)$ is the distance function, defaulting to cosine distance, $h_t$ and $\hat{h}_t$ are the features extracted from the last transformer layer of the student and teacher networks.

### 3.3.2 Contrastive Learning for Consistency

The essence of *Obs.* 2 and *Obs.* 3 lies in the inconsistency of representations during the autoregressive iterative process. Specifically, *Obs.* 2 pertains to inconsistencies between different steps in the same image, while *Obs.* 3 relates to inconsistencies between different augmented image views. Inspired by the SSL methods, we use a contrastive learning paradigm to solve such inconsistency.

Given a batch of images $\{I^{(b)}\}_{b=1}^{B}$, ST-AR applies $M$ random augmentations to each image, resulting in a set of augmented views $\{I^{(b,m)}\}_{m=1}^{M}$. These augmented images are then encoded by a VQ-GAN $q(\cdot)$, producing discrete token sequences $X \in \mathbb{Z}^{B \times M \times T}$, where $T$ denotes the token sequence length (*i.e.*, the steps of AR models). The resulting tokens are fed into both the student network $p_\theta$ and EMA teacher network $p_{\theta'}$, yielding token features $h_s = p_\theta(X) \in \mathbb{R}^{B \times M \times T \times D}$ and $h_t = p_{\theta'}(X) \in \mathbb{R}^{B \times M \times T \times D}$, where $D$ is the feature dimension. Following SimSiam[13], we employ a projector $f(\cdot)$, which consists of several MLPs, on the student features: $z_s = f(h_s)$. The projector helps prevent model collapse and enhances training stability.

To compute the contrastive loss, we randomly select $K$ token positions from the sequence length $T$, denoted as $\mathcal{I} \sim \text{RandomK}(K, T)$. The sampled features used for loss computation are:

$$\hat{z}_s = z_s[:, :, \mathcal{I}, :] \in \mathbb{R}^{B \times M \times K \times D}, \quad \hat{h}_t = h_t[:, :, \mathcal{I}, :] \in \mathbb{R}^{B \times M \times K \times D}. \tag{5}$$

We use inter-step contrastive loss $\mathcal{L}_{step}$ to enforce semantic consistency across different steps, addressing *Obs.* 2. For each sampled student feature vector $\hat{z}_s^{(b,m,i)}$, we define the positive sample as the teacher feature $\hat{h}_t^{(b,m,j)}$ extracted from the same view but a different position, while the negative samples come from other images in the batch. Formally:

$$\mathcal{L}_{step} = -\frac{1}{B} \sum_{b=1}^{B} \sum_{m=1}^{M} \sum_{i \neq j}^{K} \frac{\exp(\hat{z}_s^{(b,m,i)} \cdot \hat{h}_t^{(b,m,j)})}{\sum_{v=1}^{B} \exp(\hat{z}_s^{(b,m,i)} \cdot \hat{h}_t^{(v,m,j)})}. \tag{6}$$

In addition, we introduce inter-view contrastive loss $\mathcal{L}_{view}$ to ensure semantic consistency across different augmented views, addressing *Obs.* 3. Specifically, for a student feature $\hat{z}_s^{(b,i,k)}$, the positive sample is the teacher feature $\hat{h}_t^{(b,j,k)}$ extracted from the same token position $k$ but a different view of the same image. Negative samples come from other images in the batch. The loss is defined as:

$$\mathcal{L}_{view} = -\frac{1}{B} \sum_{b=1}^{B} \sum_{i \neq j}^{M} \sum_{k=1}^{K} \frac{\exp(\hat{z}_s^{(b,i,k)} \cdot \hat{h}_t^{(b,j,k)})}{\sum_{v=1}^{B} \exp(\hat{z}_s^{(b,i,k)} \cdot \hat{h}_t^{(v,j,k)})}. \tag{7}$$

To improve training efficiency, we set the number of image views $M = 2$ in our implementation. We conduct an ablation study about the effects of the number of different steps $K$ on generation quality, which is detailed in Table 6.

### 3.3.3 Training Losses.

We incorporate masked image modeling (Eq. 4) and contrastive learning (Eq. 6 and Eq. 7) into the conventional next-token prediction loss (Eq. 2). The final loss function can be formalized as:

$$\mathcal{L}_{ST\text{-}AR} = \mathcal{L}_{AR} + \alpha \mathcal{L}_{\text{MIM}} + \beta \frac{1}{2}(\mathcal{L}_{step} + \mathcal{L}_{view}), \tag{8}$$

where $\alpha$ and $\beta$ are the weights for the reconstruction loss and contrastive losses, respectively.

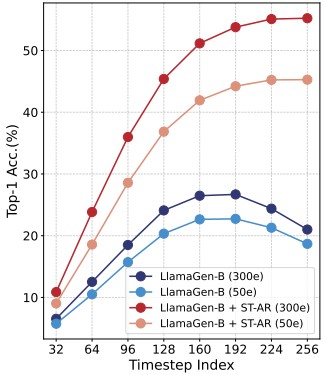

Figure 4: **Linear probing results of LlamaGen-B and our ST-AR.** Our method demonstrates consistent improvements in image understanding.

Table 1: **Comparisons between LlamaGen model and ST-AR.** All the results are evaluated without using CFG on *ImageNet*. †means the model is trained on $384 \times 384$ resolution and resized to $256 \times 256$ resolution for evaluation.

| Model | #Params | Epochs | FID↓ | sFID↓ | IS↑ | Prec.↑ | Rec.↑ |
|---|---|---|---|---|---|---|---|
| LlamaGen-B | 111M | 50 | 31.35 | 8.75 | 39.58 | 0.57 | 0.61 |
| + ST-AR | 111M | 50 | 26.58 | 7.70 | 49.91 | 0.60 | 0.62 |
| LlamaGen-B | 111M | 300 | 26.26 | 9.22 | 48.07 | 0.59 | 0.62 |
| + ST-AR | 111M | 300 | 18.44 | 6.71 | 66.18 | 0.64 | 0.62 |
| LlamaGen-L | 343M | 50 | 21.81 | 8.77 | 59.18 | 0.62 | 0.64 |
| + ST-AR | 343M | 50 | 12.59 | 6.79 | 91.19 | 0.65 | 0.64 |
| LlamaGen-L | 343M | 300 | 13.45 | 8.32 | 82.29 | 0.66 | 0.64 |
| + ST-AR | 343M | 300 | 9.38 | 6.64 | 112.71 | 0.70 | 0.65 |
| LlamaGen-XL† | 775M | 300 | 15.55 | 7.05 | 79.16 | 0.62 | **0.69** |
| LlamaGen-XXL† | 1.4B | 300 | 14.65 | 8.69 | 86.33 | 0.63 | 0.68 |
| LlamaGen-3B† | 3.1B | 300 | 9.38 | 8.24 | 112.88 | 0.69 | 0.67 |
| LlamaGen-XL | 775M | 50 | 19.42 | 8.91 | 66.20 | 0.61 | 0.67 |
| + ST-AR | 775M | 50 | 9.81 | 6.94 | 109.77 | 0.71 | 0.63 |
| + ST-AR | 775M | 300 | **6.20** | **6.47** | **147.47** | **0.73** | 0.65 |

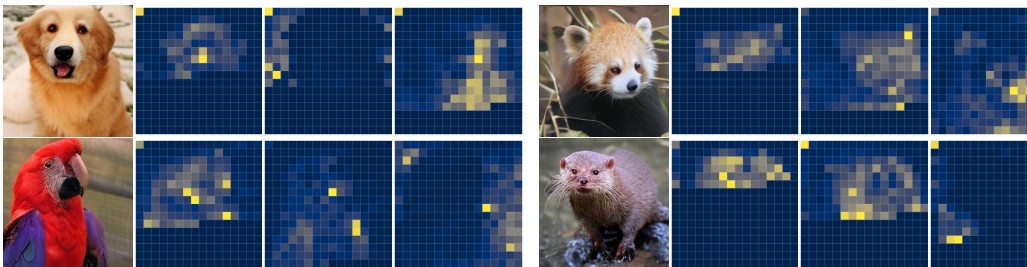

Figure 5: **Attention maps of LlamaGen-B model trained with our ST-AR method.** We utilize features from the final transformer layer, selecting random steps to draw attention maps. These maps exhibit an expanded effective receptive field, moving beyond mere focus on spatially adjacent and conditional tokens, and reveal distinct semantic patterns.

## 4 Experiments

### 4.1 Implementation Details

**Dataset.** We evaluate the effectiveness of ST-AR on the class-conditional image generation task using the widely adopted ImageNet-$256 \times 256$ dataset. We employ the same VQGAN[19] as LlamaGen[44] for tokenization, precomputing the image token sequences before training. Following LlamaGen, we also compute tokens for ten crops of the original image.

**Evaluation metrics.** Since our ST-AR generative model is trained with self-supervised losses to enhance its visual modeling capabilities, we holistically evaluate ST-AR on image understanding and generation. For image understanding, we use the top-1 accuracy of linear probing as the primary metric. We adopt the linear probing setup of MAE[25], training a linear layer for 90 epochs using the representations from the sixth layer. For image generation, we use the ADM evaluation suite and report Fréchet Inception Distance (FID)[26] as the main evaluation metric.

**Training & Inference.** All the models are trained with the same setting as LlamaGen: base learning rate of $1 \times 10^{-4}$ per 256 batch size, AdamW optimizer with $\beta_1 = 0.9, \beta_2 = 0.05$, weight decay set to 0.05 and gradient clipping set to 1.0. We train our models on images with $256 \times 256$ resolution, rather than LlamaGen with $384 \times 384$ training images. The teacher model is updated through the exponential moving average of the student model with an EMA decay of 0.9999. The class token embedding dropout ratio is 0.1 for classifier-free guidance. The contrastive loss is added on the medium of the transformer network, *i.e.* the 6-*th* layer for LlamaGen-B, 18-*th* layer for LlamaGen-L and 18-*th* layer for LlamaGen-XL. The masking ratio used for mask image modeling in Eq. 3 is set to $r = 0.25$. The number of steps used in Eq. 6 and Eq. 7 is set as $K = 4$. The weights of

Table 2: **Model comparisons on *ImageNet*-$256 \times 256$ Benchmark.** All the results are evaluated with CFG. †means the model is trained on $384 \times 384$ resolution and resized to $256 \times 256$ resolution for evaluation. ST-AR consistently beats baseline LlamaGen on all model sizes and training costs.

| Type | Model | #Params | Epochs | FID↓ | sFID↓ | IS↑ | Prec.↑ | Rec.↑ |
|---|---|---|---|---|---|---|---|---|
| GAN | BigGAN [7] | 112M | | 6.95 | 7.36 | 171.40 | 0.87 | 0.28 |
| | StyleGan-XL [42] | 166M | | 2.30 | 4.02 | 265.12 | 0.78 | 0.53 |
| Diff. | LDM-4[41] | 400M | | 3.60 | 5.12 | 247.67 | 0.87 | 0.48 |
| | DiT-XL[37] | 675M | 1400 | 2.27 | 4.60 | 278.24 | 0.83 | 0.57 |
| | SiT-XL[34] | 675M | 1400 | 2.15 | 4.50 | 258.09 | 0.81 | 0.60 |
| Masked AR | MaskGIT[11] | 227M | 300 | 6.18 | - | 182.10 | 0.80 | 0.51 |
| | MaskGIT-re[11] | 227M | 300 | 4.02 | - | 355.60 | 0.83 | 0.50 |
| Parallelized AR | VAR-$d$16[46] | 310M | | 3.30 | - | 274.4 | 0.84 | 0.51 |
| | VAR-$d$20[46] | 600M | | 2.57 | - | 302.6 | 0.83 | 0.56 |
| Casual AR | VQGAN[19] | 1.4B | | 15.78 | - | 74.30 | - | - |
| | VQGAN-re[19] | 1.4B | | 5.20 | - | 280.30 | - | - |
| | LlamaGen-B† [44] | 111M | 300 | 6.09 | 7.24 | 182.54 | 0.85 | 0.42 |
| | LlamaGen-L† [44] | 343M | 300 | 3.08 | 6.09 | 256.07 | 0.83 | 0.52 |
| | LlamaGen-XL† [44] | 775M | 300 | 2.63 | 5.59 | 244.09 | 0.81 | 0.58 |
| Casual AR | LlamaGen-B | 111M | 300 | 5.46 | 7.50 | 193.61 | 0.84 | 0.46 |
| | **+ ST-AR** | 111M | 300 | 4.09 | 6.72 | 246.29 | 0.86 | 0.47 |
| | LlamaGen-L | 343M | 300 | 3.81 | 8.49 | 248.28 | 0.83 | 0.52 |
| | **+ ST-AR** | 343M | 300 | 2.98 | 6.44 | 264.11 | 0.85 | 0.53 |
| | LlamaGen-XL | 775M | 50 | 3.39 | 7.02 | 227.08 | 0.81 | 0.54 |
| | **+ ST-AR** | 775M | 50 | 2.72 | 6.03 | 254.59 | 0.83 | 0.57 |
| | **+ ST-AR** | 775M | 300 | 2.37 | 6.05 | 270.59 | 0.82 | 0.58 |

Table 3: **The effects of proposed losses.** ST-AR improves linear probing and generation quality.

| Model | $\mathcal{L}_{\text{MIM}}$ | $\mathcal{L}_{step}$ | $\mathcal{L}_{view}$ | FID↓ | sFID↓ | IS↑ | Prec.↑ | Rec.↑ | LP Acc.(%)↑ |
|---|---|---|---|---|---|---|---|---|---|
| LlamaGen-B | | | | 31.35 | 8.75 | 39.58 | 0.57 | 0.61 | 18.68 |
| | √ | | | 30.58 | 8.94 | 41.95 | 0.59 | 0.59 | 22.71 |
| | √ | √ | | 28.02 | 8.21 | 46.20 | 0.59 | 0.61 | 27.73 |
| | √ | | √ | 27.78 | **7.52** | 45.88 | **0.60** | 0.61 | 38.31 |
| **+ ST-AR (Ours)** | √ | √ | √ | **26.58** | 7.70 | **49.91** | **0.60** | 0.62 | **45.27** |

reconstruction loss and contrastive loss in Eq. 8 are set to $\alpha = 1.0$ and $\beta = 0.5$ by default. For inference, we use the same sampling strategy as LlamaGen.

## 4.2  Main Results

**Image understanding.** The linear probing results are shown in Figure 4. ST-AR significantly enhances the linear probing performance of the baseline model, LlamaGen-B, across all steps, demonstrating improved image understanding capabilities. Importantly, the accuracy does not degrade after the $192\text{-}th$ step, indicating that LlamaGen trained with ST-AR effectively preserves semantic information from previous iterations during the sampling process.

In Figure 5, we visualize the attention maps of the last layer at different steps. Compared to the baseline model (Figure 2), ST-AR not only significantly expands the scope of attention but also focuses on semantically relevant regions, further demonstrating that ST-AR effectively enhances the learning of visual semantic representations.

**Class-conditional image generation.** As previously stated, the enhancement in image understanding also leads to higher generation quality. We first compare the LlamaGen models trained with ST-AR to their vanilla counterparts. As shown in Table 1, ST-AR achieves significant performance improvements across all LlamaGen variants. Specifically, for LlamaGen-XL, training with ST-AR for 50 epochs improves the FID score by approximately 10, reducing it from $19.42$ to $9.81$ compared to the vanilla counterpart. Further training for 300 epochs leads to an FID of $6.20$, which is even stronger than LlamaGen-3B with $4\times$ parameters.

In Table 2, we provide results using classifier-free guidance (CFG) and comparisons with methods from other paradigms, including GANs, diffusion models, masked AR, and parallelized AR. ST-AR

Table 4: Ablation on mask ratio.

| Ratio | FID↓ | sFID↓ | IS↑ |
|---|---|---|---|
| 0.15 | 28.62 | **7.28** | 44.58 |
| 0.25 | 26.58 | 7.70 | **49.91** |
| 0.35 | **26.36** | 8.20 | 49.73 |
| 0.45 | 27.50 | 8.31 | 47.15 |

Table 5: Ablation on contrastive loss depth.

| Depth | FID↓ | sFID↓ | IS↑ |
|---|---|---|---|
| 3 (1/4-d) | 27.34 | **7.49** | 46.23 |
| 6 (1/2-d) | **26.58** | 7.70 | **49.91** |
| 9 (3/4-d) | 28.76 | 8.66 | 44.73 |
| 12 (1-d) | 29.45 | 8.56 | 43.32 |

Table 6: Ablation on the number of selected steps.

| #Steps | FID↓ | sFID↓ | IS↑ |
|---|---|---|---|
| 2 | 27.50 | 8.31 | 47.15 |
| 4 | 26.58 | 7.70 | 49.91 |
| 8 | 26.54 | **7.61** | 48.70 |
| 16 | **25.78** | 7.86 | **50.66** |

achieves consistent and significant improvements over LlamaGen while also delivering performance comparable to other state-of-the-art methods.

**Qualitative Comparison.** In Figure 6, we generate images using the same random seed for LlamaGen-B and LlamaGen-B + ST-AR. It can be observed that, due to the lack of global and semantic information, images generated by LlamaGen exhibit distortions and object discontinuities, while images generated with ST-AR appear more natural. This also highlights the importance of incorporating high-level semantic information.

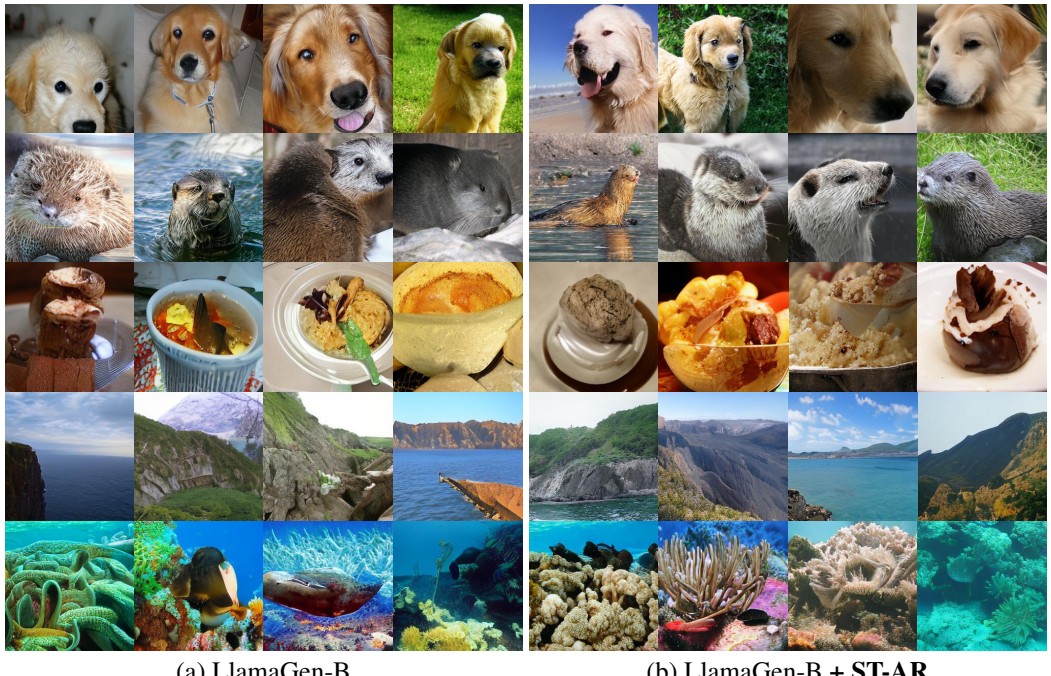

(a) LlamaGen-B  (b) LlamaGen-B + **ST-AR**

Figure 6: **Qualitative comparison between (a) LlamaGen-B and (b) LlamaGen-B + ST-AR.**

## 4.3 Ablation Studies

We conduct comprehensive experiments on different configurations of ST-AR. All reported results are obtained using LlamaGen-B model trained for 50 epochs.

**Effectiveness of Training Losses.** We conduct experiments to validate the effectiveness of the three loss functions in ST-AR, namely $\mathcal{L}_{\text{MIM}}$, $\mathcal{L}_{step}$, and $\mathcal{L}_{view}$. The results are shown in Table 3. All three losses improve linear probing accuracy, thereby enhancing generation quality. Among them, the inter-view contrastive loss $\mathcal{L}_{view}$ contributes more to the improvement in linear probing accuracy compared to the inter-step contrastive loss $\mathcal{L}_{step}$. Notably, equipping LlamaGen-B with all three losses significantly increases its linear probing accuracy from $18.68\%$ to $45.27\%$.

In Figure 7, we visualize the attention maps of models trained with each of the three losses individually. The MIM loss significantly expands the attention range but fails to capture semantic information, consistent with the results in Table 3. Both the inter-step contrastive loss and inter-view contrastive loss slightly expand the attention range but significantly highlight semantically relevant areas.

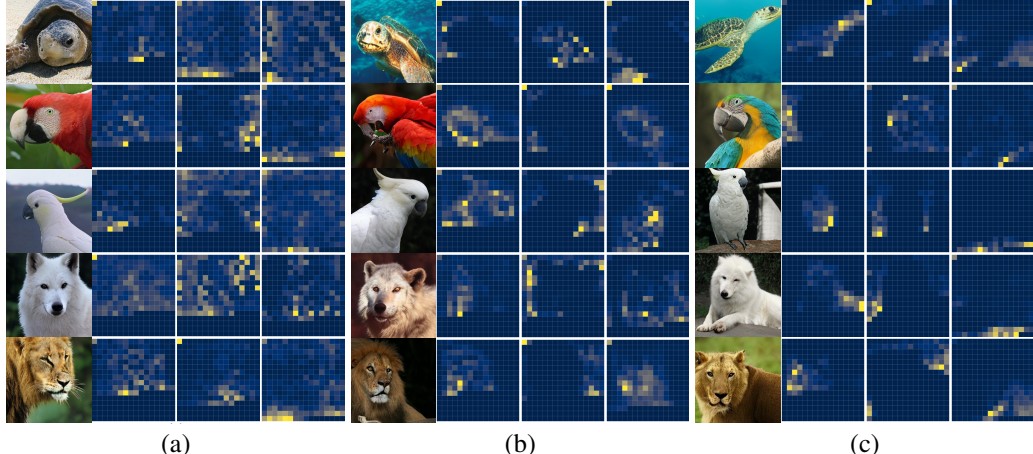

Figure 7: **Attention maps of LlamaGen-B trained with (a) MIM loss, (b) inter-step contrastive loss, and (c) inter-view contrastive loss individually.**

**Effect of Mask Ratio.**    Masked image modeling is a key design in ST-AR, as discussed in Section 3.3.1, it expands the effective receptive field of the network. In Table 4, we examine the effect of the mask ratio on generation performance. The FID score is lowest when the mask ratio is $0.35$. However, increasing the mask ratio leads to degradation in sFID, indicating that masking too many tokens can negatively affect the learning of low-level spatial structures.

**Effect of Contrastive Loss Depth.**    We validate the impact of incorporating the two contrastive losses, $\mathcal{L}_{step}$ and $\mathcal{L}_{view}$, at different depths of the network. There has long been a view that image generators consist of an encoder and a decoder. The results shown in Table 5 align with this perspective, demonstrating that applying contrastive losses at the $6 - th$ layer (half the depth) yields the best performance.

**Effect of the Number of Steps.**    As described in Section 3.3.2, we randomly select $K$ different steps for contrastive learning. In Table 6, we examine the impact of the number of steps $K$. Larger values of $K$ lead to better generation performance. However, the improvement becomes marginal for $K > 4$. Therefore, we set $K = 4$ by default.

## 5   Conclusion

In this work, we focus on investigating the visual understanding capabilities of autoregressive models for image generation, offering an in-depth analysis and identifying three fundamental challenges that hinder the learning of high-level visual semantics. We demonstrate that these challenges can be effectively addressed by incorporating representation learning objectives, leading to a novel training framework: Self-guided Training for AutoRegressive models (ST-AR). ST-AR employs masked image modeling to broaden attention regions while utilizing contrastive learning to maintain semantic consistency across steps and views. Extensive experiments validate ST-AR's effectiveness in enhancing visual understanding, which consequently improves image generation quality.

**Limitations & societal impacts.** The main limitation of this work lies in increased training costs, which we will address in future research. While ST-AR establishes a novel training paradigm for autoregressive image generation with potential industry applications, it may also raise concerns regarding image manipulation risks.

## Acknowledgements

This work was supported by the JC STEM Lab of AI for Science and Engineering, funded by The Hong Kong Jockey Club Charities Trust, the Research Grants Council of Hong Kong (Project No. CUHK14213224).

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
