# OpenReview forum: "Understand Before You Generate: Self-Guided Training for Autoregressive Image Generation"
_NeurIPS.cc/2025/Conference — NeurIPS 2025 poster_

### Official Review · Reviewer_qqSW · 2025-07-01

**Clarity:** 4
**Significance:** 4
**Originality:** 4
**Rating:** 5
**Confidence:** 5

**Summary:**

This paper investigates the limitations of autoregressive (AR) models in image generation, particularly their poor visual understanding (based on LlamaGen). The authors identify three key issues: (1) over-reliance on local and conditional information (each next token relies on only spatial neighbors), (2) semantic inconsistency across generation steps (seeing all tokens have lower linear probe accuracy than seeing partialy), and (3) lack of spatial invariance in visual tokens (e.g., VQGAN encodes differently if image is shifted). To address these, they propose ST-AR (Self-guided Training for AR), a training framework that incorporates masked attention modeling and two contrastive learning objectives (inter-step and inter-view). These augment the standard AR next-token loss without modifying inference. ST-AR significantly improves both visual representation quality (e.g., linear probing accuracy from 21% to 55%) and generation quality (e.g., up to 49% FID improvement), all without relying on external pretrained models.

**Questions:**

1. I might have missed this information from the paper, can you quantify the training overhead introduced by ST-AR?
2. Since contrastive loss is applied, what is the effect of batch size?
3. Can you discuss how ST-AR be extended to other AR domains like 3d or video?
4. I notice the high norm issue on the first patch, would adding register tokens help? (Also, whether adding this register token have any new insight?)

**Ethical Concerns:**

["NO or VERY MINOR ethics concerns only"]

**Final Justification:**

thanks authors for their rebuttal. concern addressed and i am keeping my score as Accept.

**Limitations:**

yes. computation cost but it is unavoidable.

**Quality:**

4

**Strengths And Weaknesses:**

Quality:
- The paper is technically sound, well-formulated, and experimentally robust. Very insightful analysis. The integration of masked attention and contrastive learning into AR training is effective and carefully designed. The experiments on ImageNet are extensive, showing strong improvements across multiple AR model sizes and metrics, with clear ablation studies validating each component.

Clarity:
- The writing is clear and well-organized. Visualizations and architecture diagrams support understanding. The authors provide precise explanations of why applying attention masking and contrastive objectives.

Originality:
- While the self-supervised tools used (e.g., contrastive loss, masked modeling) are established, their application to AR image generation is novel and well-motivated. The diagnosis of AR's semantic weaknesses is insightful and targeted design of ST-AR is a key original contribution.

Significance:
- This work has broad implications for scaling AR models across modalities. Representation learning is always the key to better model, by improving representation learning within AR models, ST-AR makes them more viable competitors to diffusion models, especially given the efficiency and modularity of AR transformers.

Weaknesses:

1. ST-AR increases training complexity and compute cost (EMA teacher, additional losses).
2. The careful composition of known components is a bit trivial and straightforward, although effective.
3. Text-to-image generation experiment detail in appendix is not clear enough and perhaps can move some results to main text.
4. Fig. 3 can be improved, the loss parts can't be understood easily without reading the text.

---

> ### Author Rebuttal · Authors · 2025-07-31
>
> We truly thank the reviewer for your detailed review and thoughtful comments in helping us enhance the quality of our paper. Below, we provide detailed point-to-point responses addressing each of your valuable suggestions:
>
> **Q1: Training overhead of ST-AR.**
>
> In Table D1, we quantify the training overhead introduced by ST-AR. All experiments are conducted using 8 NVIDIA A100 GPUs (80GB each), with a global batch size of 256 and an input image resolution of $256 \times 256$. We report the average training time per iteration as well as the peak memory consumption during training.
> The training overhead of ST-AR is higher than the baseline LlamaGen but comparable to that of REPA. However, ST-AR achieves significantly better performance gains. Reducing the training cost of ST-AR will be a key direction for our future work.
>
> **Table D1: Comparison of training costs.**
> |Model|Epochs|FID|sFID|IS|Prec.|Rec.|Training Latency (s/iter)|Peak GPU Memory (GB)|
> |-|-|-|-|-|-|-|-|-|
> |LlamaGen-B|50|31.35|8.75|39.58|0.57|0.61|**0.0493**|**~7**|
> |LlamaGen-B + REPA|50|28.50|7.78|44.72|0.59|**0.62**|0.0658|~8|
> |LlamaGen-B + ST-AR|50|**26.58**|**7.70**|**49.91**|**0.60**|**0.62**|0.0698|~10|
>
> **Q2: Effect of Batch Size**
>
> We evaluate the impact of batch size on generation performance and representation quality. As shown in Table D2, a larger batch size is more beneficial for semantic information learning, resulting in higher linear probing accuracy. However, in terms of generation quality, a batch size of 512 performs worse than 256. We attribute this to the reduced number of training iterations when using a larger batch size.
>
>
> **Table D2: Effect of batch size.**
> |Batch size|FID|sFID|IS|Prec.|Rec.|LP Acc.(%)|
> |-|-|-|-|-|-|-|
> |128|28.83|9.46|46.47|**0.60**|0.61|45.03|
> |256|**26.58**|**7.70**|**49.91**|**0.60**|**0.62**|45.27|
> |384|27.70|8.73|47.25|0.59|**0.62**|45.81|
> |512|29.29|8.49|45.38|0.57|**0.62**|**46.13**|
>
> **Q3: Extending ST-AR to other AR domains.**
>
> One of the key contributions of our work is establishing a methodology for analyzing autoregressive generation models. By quantifying the scope of the attention map and evaluating linear probing performance, we gain insights into the intrinsic mechanisms of autoregressive models.
>
> To extend ST-AR to other domains, such as video generation, conducting similar analyses is a crucial first step. Intuitively, for video generation, we would need to further examine the attention weights between frames and assess the semantic consistency across frames. Based on these insights, we could then introduce additional high-level inter-frame representation alignment losses to ST-AR, aiming to enhance the video generation performance of baseline models.
>
> **Q4: High norm issue on the first patch**
>
> The input at the first time step is a conditional vector that guides the generation process. It is reasonable for this vector to always have a high norm.
>
> To validate the effect of register tokens, we inserted 4 register tokens between the first token (conditional vector) and the second token (the first image patch). The image generation process starts from the last register token.
> As shown in Table D3, the introduction of register tokens leads to a slight performance improvement. However, visualizations of the attention maps reveal that the conditional vector still exhibits a high norm.
>
> **Table D3: Effect of register tokens.**
> ||FID|sFID|IS|Prec.|Rec.|
> |-|-|-|-|-|-|
> |w/o registers|26.58|7.70|49.91|**0.60**|0.62|
> |w/ registers|**25.98**|**7.69**|**50.78**|**0.60**|**0.63**|

---

> ### Author Response · Authors · 2025-08-06
>
> Thank you for reviewing our work and providing positive feedback. We are glad to address any further questions or comments.

---

> > ### Comment · Reviewer_qqSW · 2025-08-06
> >
> > thanks authors for their rebuttal. concern addressed and i am keeping my score as Accept.

---

### Official Review · Reviewer_gtuF · 2025-07-02

**Clarity:** 3
**Significance:** 3
**Originality:** 3
**Rating:** 4
**Confidence:** 4

**Summary:**

This paper investigates the underlying limitations of the standard next-token prediction paradigm for autoregressive (AR) image generation. The authors systematically identify three key properties that hinder the learning of high-level visual semantics: (1) over-reliance on local and conditional dependence, (2) semantic inconsistency across generation steps, and (3) a lack of spatial invariance in the visual tokens.
To address these issues, the paper proposes ST-AR (Self-guided Training for AutoRegressive models), a novel training framework that integrates self-supervised learning objectives directly into the AR training process. Specifically, ST-AR employs masked image modeling on attention maps to encourage a wider receptive field and uses two forms of contrastive learning (inter-step and inter-view) to enforce semantic consistency. A key advantage of this approach is that it enhances the model's understanding and generation quality without requiring pre-trained representation models or altering the standard AR sampling strategy at inference time. Experiments on ImageNet show that ST-AR significantly improves both image understanding (linear probing accuracy) and generation quality (FID scores) across various model scales.

**Questions:**

1. In the final loss function (Eq. 8), the weights for the new loss terms are set to α = 1.0 and β = 0.5. How sensitive is the model's performance to these specific hyperparameter choices? Was a sweep performed to find these values, and could you provide some intuition on how the balance between the standard AR loss and the new self-supervised losses affects the training dynamic?

2. The framework combines a reconstruction-style objective (L_MIM) with consistency-based contrastive objectives (L_step, L_view). Did you observe any interesting interplay or potential conflict between these losses during training? For example, does the MIM objective, which forces the model to use broader context, directly aid the inter-step consistency objective, or are they largely independent mechanisms for improvement?

3. The contrastive losses are applied at a middle layer of the network (e.g., the 6th layer for LlamaGen-B). The ablation in Table 5 shows this is optimal. Do you have any intuition for why this "middle-ground" is more effective than applying the constraint on earlier (more textural) or later (more semantic) features? Does this suggest that ST-AR is primarily helping to structure the mid-level representations of the network?

**Ethical Concerns:**

["NO or VERY MINOR ethics concerns only"]

**Limitations:**

Yes.

**Paper Formatting Concerns:**

No.

**Quality:**

3

**Strengths And Weaknesses:**

Strength

1. This work systematically investigate into why autoregressive models struggle with visual semantics. The authors go beyond simply proposing a new method and provide a clear, well-supported diagnosis of the core problems. This principled analysis provides a strong foundation and clear motivation for the proposed ST-AR framework.

2. The proposed ST-AR framework is a novel and clever integration of self-supervised learning techniques (MIM and contrastive learning) into the autoregressive training loop.

Weakness

1. The authors correctly identify "increased training costs" as a limitation in the conclusion. However, this is a significant practical concern that warrants a more detailed discussion. The ST-AR framework introduces a teacher network, requires processing multiple views of each image, and computes three additional loss terms. This will substantially increase both the computational and memory requirements for training compared to the baseline.

2. This paper positions itself as an approach that avoids reliance on pre-trained representation models (a key advantage). However, a stronger argument could be made by directly discussing or comparing against methods that do leverage such models (e.g., by using a pre-trained VAE or distilling knowledge from a model like DINO/CLIP). A brief discussion on the conceptual pros and cons (e.g., ST-AR's "self-guided" nature vs. the potential power of a large-scale pre-trained teacher) would better situate the work within the broader landscape of improving semantic understanding in generative models.

---

> ### Author Rebuttal · Authors · 2025-07-31
>
> We truly thank the reviewer for your detailed review and thoughtful comments in helping us enhance the quality of our paper. Below, we provide detailed point-to-point responses addressing each of your valuable suggestions:
>
> **Q1: Training cost of ST-AR.**
>
> In Table C1, we provide a fair comparison of the training costs for LlamaGen, REPA [1], and ST-AR.
> All experiments are conducted under the same training settings, using 8 A100 GPUs with a global batch size of $256$ and an input image resolution of $256\times256$. We report the average runtime per training iteration and peak memory usage.
> Compared to the baseline LlamaGen and the REPA (which requires a pre-trained image encoder), ST-AR incurs slightly higher training costs but delivers significant performance improvements.
> Reducing the training cost of ST-AR will be an important focus of our future work.
>
> **Table C1: Comparison of training costs and comparison between ST-AR and pretraining-based methods.**
> |Model|Pretrained Encoder|Epochs|FID|sFID|IS|Prec.|Rec.|Training Latency (s/iter)|Peak GPU Memory (GB)|
> |-|-|-|-|-|-|-|-|-|-|
> |LlamaGen-B||50|31.35|8.75|39.58|0.57|0.61|**0.0493**|**~7**|
> |LlamaGen-B + REPA|DINOv2-B|50|28.50|7.78|44.72|0.59|**0.62**|0.0658|~8|
> ||MAE-B|50|31.27|9.12|40.27|0.58|0.60|0.0662|~8|
> |LlamaGen-B + ST-AR||50|**26.58**|**7.70**|**49.91**|**0.60**|**0.62**|0.0698|~10|
>
> **Q2: Comparison with methods requiring a pre-trained image encoder.**
>
> We follow REPA[1] to align the intermediate features of LlamaGen-B with the representations from DINOv2-B and MAE-B, respectively.
> As shown in Table C1, LlamaGen-B with REPA demonstrates better generation performance compared to the baseline but falls short of ST-AR.
>
> We analyze the pros and cons of ST-AR and pretraining-based methods from two perspectives:
> 1. **Spatial misalignment when aligning intermediate features from autoregressive models with pre-trained visual representations.** For position $i$, an autoregressive model involves two adjacent tokens: the $(i-1)$-th token as the input and the $i$-th token as the predicted output.
> In contrast, visual encoders like DINOv2 extract features from the $i$-th image patch.
> Forcing alignment may introduce inductive bias into the autoregressive model. In comparison, ST-AR does not rely on pretrained visual encoders, thereby avoiding the spatial misalignment.
> 2. **Representation quality.** While visual encoders pretrained on larger-scale datasets may learn better semantic representations, these representations do not necessarily contribute to learning for image generation. In contrast, ST-AR unifies representation learning and image generation, allowing it to directly learn features that are more beneficial for autoregressive image generation.
>
> **Q3: The effect of $\alpha$ and $\beta$ and the balance between AR loss and self-supervised losses.**
>
> In Table C2, we examine the effect of $\alpha$ and $\beta$ on generative performance.
> When training for 50 epochs:
> - ST-AR performs best when $\alpha=1.0$ and $\beta=0.5$.
> - $\alpha$ is the weight of the MIM loss. Reducing the value of $\alpha$ decreases the model's effective receptive field, leading to poorer generation quality.
> - $\beta$ is the weight of the two contrastive losses. A lower value ($0.25$) makes it difficult to learn semantic information, while values greater than $0.5$ might hinder the learning of the autoregressive loss.
>
> For training dynamics, we evaluate the performance of ST-AR with different $\alpha$ and $\beta$ at $20$ epochs and $50$ epochs. As in Table C2, the best performance is achieved by set $\alpha=1.0$ and $\beta=1.0$ at $20$ epochs. This indicates that larger $\alpha$ and $\beta$ contribute to learning better semantic representations, which accelerates the convergence of the autoregressive model. However, larger contrastive loss may hinder the learning of autoregressive loss, leading to inferior performance of setting $\{\alpha=1.0, \beta=1.0\}$ at $50$ epochs.
>
> **Table C2: Ablation study for $\alpha$ and $\beta$.**
> |Epochs|$\alpha$|$\beta$|FID|sFID|IS|Prec.|Rec.|
> |-|-|-|-|-|-|-|-|
> |50|1.00|0.25|26.72|8.17|48.55|**0.61**|0.61|
> ||1.00|0.50|**26.58**|**7.70**|**49.91**|0.60|**0.62**|
> ||1.00|0.75|26.67|7.71|48.41|0.60|0.61|
> ||1.00|1.00|27.32|7.86|47.59|0.59|0.61|
> ||0.75|0.50|27.07|**7.70**|47.56|**0.61**|0.61|
> ||0.50|0.50|27.57|7.71|46.70|**0.61**|0.59|
> ||0.25|0.50|27.92|8.08|44.54|**0.61**|0.59|
> |20|1.00|0.25|38.09|10.28|33.83|0.52|0.61|
> ||1.00|0.50|36.88|9.12|35.38|0.52|0.62|
> ||1.00|0.75|36.28|9.67|36.27|**0.54**|0.61|
> ||1.00|1.00|**35.27**|**8.45**|**36.79**|0.53|**0.63**|
> ||0.75|0.50|38.35|10.42|35.05|0.52|0.62|
> ||0.50|0.50|38.63|9.96|34.54|0.52|0.62|
> ||0.25|0.50|39.31|10.26|33.66|0.51|0.62|
>
> **Q4: The interaction between the reconstruction objective and the contrastive objectives.**
>
> The interaction between the reconstruction objective and contrastive objectives exhibits both conflict and mutual benefits.
>
> **Conflict.** MIM and contrastive learning capture different levels of semantic information, resulting in a unique challenge of representation conflict. The inter-step and inter-view contrastive losses are designed to address semantic inconsistencies across steps and views, respectively, promoting the learning of global, image-level semantics. In contrast, the MIM loss trains the autoregressive (AR) model by reconstructing the correct semantics of masked tokens, emphasizing token-level semantics. To mitigate this conflict, we compute the contrastive losses on intermediate-layer features and the MIM loss on final-layer features. As in the Table 5 of our manuscript, this design can bring the best performance.
>
> **Mutual Benefits.** As shown in Figure A1 of the supplementary material, contrastive objectives face limitations in expanding the effective receptive field, which restricts the learning of wide-range high-level semantics (e.g., objects). The reconstruction objective effectively addresses this issue. On the other hand, the reconstruction objective alone struggles to capture precise semantic entities, whereas the contrastive objectives complement and expand the capabilities of MIM, leading to more robust learning of semantic representations.
>
> **Q5: Applying contrastive losses to the middle layer.**
>
> The contrastive losses in ST-AR aim to align high-level semantic information across different timesteps and views. Notably, the most semantically meaningful features exist in the middle layer of the autoregressive model.
>
> The autoregressive model processes preceding tokens as input to predict the next token, which can conceptually be divided into two stages:
> 1. The shallow subnetwork maps the preceding tokens into a high-level representation.
> 2. The deeper subnetwork decodes this high-level representation into the next token.
>
> The most semantically meaningful features are produced in the middle layer rather than the final layer. This claim is supported by the linear probing results shown in Table C3.
>
> **Table C3: Linear probing results of different layers in LlamaGen-B.**
> |Layer Index|3|6|9|12|
> |-|-|-|-|-|
> |LP Acc.(%)|8.42|**18.68**|16.90|12.39|
>
> [1] Representation Alignment for Generation: Training Diffusion Transformers Is Easier Than You Think. ICLR 2025

---

> ### Author Response · Authors · 2025-08-06
>
> Thank you for your effort in reviewing our work and for your positive feedback. We are happy to address any further questions or comments.

---

### Official Review · Reviewer_Paj7 · 2025-07-03

**Clarity:** 3
**Significance:** 2
**Originality:** 2
**Rating:** 4
**Confidence:** 4

**Summary:**

This paper investigates the challenges in visual understanding for autoregressive (AR) image generation models and proposes a self-guided training framework, ST-AR (Self-guided Training for AutoRegressive models), to address these issues. The authors identify three key limitations in AR models:
1. Local and conditional dependence: AR models overly rely on initial conditioning tokens and spatially adjacent information, as shown by attention map analyses.
2. Inter-step semantic inconsistency: Semantic information varies across different generation timesteps, leading to degraded global modeling.
3. Spatial invariance deficiency: Visual tokenizers lack invariance, causing ambiguous tokenization for slightly perturbed images.

ST-AR integrates self-supervised learning techniques to enhance AR models without pre-trained representations:
- **Masked image modeling (MIM)** on attention maps to expand the receptive field.
- **Inter-step contrastive loss** to ensure semantic consistency across timesteps.
- **Inter-view contrastive loss** to align representations from different image augmentations.

**Questions:**

refer to the weakness

**Ethical Concerns:**

["NO or VERY MINOR ethics concerns only"]

**Final Justification:**

Thank you for the author's reply, which has basically solved my problem. I will raise the rating as accept.

**Paper Formatting Concerns:**

For me, this paper is very well-written and there are no obvious formatting problems.

**Quality:**

2

**Strengths And Weaknesses:**

Strengths:

1/ The research motivation of this work is to enhance the performance of auto-regressive models in understanding tasks. The authors found that next token prediction-based AR models have obvious defects in understanding tasks (Autoregressive models primarily rely on local and conditional information, Causal Attention Challenges Bi-directional Image Context Modeling). Based on these findings, the authors introduced methods commonly used in traditional visual representation learning, such as MIM and contrastive learning, which can effectively improve the performance of ImageNet class-conditional generation and downstream understanding tasks, such as ImageNet linear probing.

2/ The authors conducted detailed experiments. In Table 1 & Table 2, the performance has significantly improved compared with the LlamaGen baseline. This method of adding additional constraints to align representations is similar in spirit to REPA and VAVAE( Reconstruction vs. Generation: Taming Optimization Dilemma in Latent Diffusion Models), and it indeed can effectively improve the convergence speed and effect. However, I wonder whether this will make the model fall into a local optimum, because the model needs to learn both representation and generation during training. My personal view is that aligning to a pre-trained representation is more stable for optimization.

Weaknesses:

1/ Personally, I think this work somewhat resembles stitching together multiple established approaches that are known to work—such as MIM and contrastive learning, which are effective for representation learning. As such, I believe the work lacks substantial innovation.

2/ I am skeptical about the upper limits of pure visual representation learning. First, the validation of understanding tasks should be conducted on a broader range of multimodal tasks and downstream visual tasks. Currently, the effectiveness of visual encoders is mostly verified for downstream MLLM (Multimodal Large Language Model) tasks like VQA, grounding, and reasoning. However, in such MLLM tasks, approaches like MAE and contrastive learning have been abandoned in favor of semantic-aligned models such as CLIP and the SigLIP series. Therefore, my view is that integrating MIM and contrastive learning into autoregressive models does not represent the correct path.

---

> ### Author Rebuttal · Authors · 2025-07-31
>
> We truly thank the reviewer for your detailed review and thoughtful comments in helping us enhance the quality of our paper. Below, we provide detailed point-to-point responses addressing each of your valuable suggestions:
>
> **Q1: Innovations of ST-AR**
>
> To the best of our knowledge, we are the first to explore the representation learning capabilities of AR models and to identify the challenges of transferring next-token prediction to the visual domain.
> The innovations of ST-AR can be summarized in two main aspects:
> - We introduce an effective methodology to analyze and identify the limitations of autoregressive models in visual representation learning by evaluating attention maps and linear probing performance.
> - We demonstrate that by appropriately incorporating well-established self-supervised learning techniques, the representation quality of AR models can be significantly improved, which subsequently enhances generation performance.
>
> Furthermore, the application of established self-supervised approaches to AR generation has not been explored before. ST-AR is the first unified framework that bridges representation learning and AR image generation.
>
> Notably, ST-AR outperforms methods that align AR features with pre-trained representations (e.g., REPA). In Table C1, we align the intermediate features of LlamaGen-B to those of either DINOv2-B or MAE-B, but both approaches perform worse than ST-AR. We attribute this to two reasons:
> - **Spatial misalignment between AR features and pre-trained representations.** In AR models, each time step uses the preceding token as input and predicts the next token as output, while pre-trained visual encoders focus on a single token. Directly aligning representations from these two paradigms is suboptimal. For more details, please refer to our response to **Reviewer rKee Q2**.
> - **Uncertainty in desired representations for AR models.** Different pre-trained visual encoders learn different types of representations, and it remains unclear what specific kinds of representations are optimal for AR models. ST-AR avoids introducing inductive bias by employing a self-guided approach that jointly trains representations and generation within a unified framework.
>
> **Table C1: Comparison between ST-AR and pretraining-based methods.**
> |Model|Pretrained Encoder|Epochs|FID|sFID|IS|Prec.|Rec.|
> |-|-|-|-|-|-|-|-|
> |LlamaGen-B||50|31.35|8.75|39.58|0.57|0.61|
> |LlamaGen-B + REPA|DINOv2-B|50|28.50|7.78|44.72|0.59|
> ||MAE-B|50|31.27|9.12|40.27|0.58|0.60|
> |LlamaGen-B + ST-AR||50|**26.58**|**7.70**|**49.91**|**0.60**|**0.62**|
>
> **Q2: The importance of pure visual representation learning.**
>
> The effectiveness of incorporating high-level visual representations into generative models has been well demonstrated[1][2].  Notably, REPA conducts the comparison between CLIP features and pure visual representations (e.g., DINOv2), showing that DINOv2 provides a stronger boost to generation performance than CLIP. **This suggests that the potential of pure visual representation learning is not inferior to vision-language alignment.**
>
> Pure visual representation learning and vision-language alignment (e.g., CLIP) each have their own strengths and weaknesses, and they can be combined to obtain richer and more comprehensive representations. Enhancing image understanding through pure visual representation learning remains necessary and valuable.
> - **Representation quality.** Pure visual representation learning excels at capturing high-quality features of diverse objects. For example, DINO features have been effectively utilized for unsupervised instance segmentation[3][4]. In contrast, CLIP focuses on aligning image content with text embeddings, which can sometimes result in slightly inferior features.
> - **Data Requirements.** Pure visual representation learning requires only a large collection of images as training data, making it less resource-intensive in comparison to vision-language alignment methods like CLIP, which depend on costly textual annotations paired with images.
>
> In practice, combining pure visual representation learning with vision-language alignment can significantly enhance visual understanding at various levels. For instance, SPHINX-X[5] leverages both DINOv2 and CLIP as its visual encoders to achieve superior performance.
>
>
> *Reference*
>
> [1] Representation Alignment for Generation: Training Diffusion Transformers Is Easier Than You Think, ICLR 2025
>
> [2] REPA-E: Unlocking VAE for End-to-End Tuning with Latent Diffusion Transformers, ICCV 2025
>
> [3] Cut and Learn for Unsupervised Object Detection and Instance Segmentation, CVPR 2023
>
> [4] Freesolo: Learning to segment objects without annotations. CVPR 2022
>
> [5] SPHINX-X: Scaling Data and Parameters for a Family of Multi-modal Large Language Models. ICML 2024

---

> > ### Comment · Reviewer_Paj7 · 2025-08-05
> >
> > * Thank you for your reply and supplementary experiments. Judging from the results, it is indeed more reasonable for ST-AR to align with the DINO feature space compared to Repa's direct alignment.
> >
> > * I also agree that visual representation alignment will be beneficial to generation, and I recognize the technical contribution of this work.

---

> > > ### Author Response · Authors · 2025-08-06
> > >
> > > Thank you for recognizing the contribution of our work and acknowledging that we have addressed most of your concerns.
> > >
> > > However, we would like to make the following clarifications:
> > > - By accurately identifying the shortcomings of autoregressive models in visual representation learning and introducing targeted self-supervised losses, **our proposed training framework, ST-AR, does not rely on pretrained vision encoders such as DINO.**
> > > - In the rebuttal, we further demonstrate that, for autoregressive models, **aligning with pretrained representations (e.g., DINOv2 and MAE) is less effective compared to the end-to-end joint training of ST-AR.**
> > >
> > > If we have fully addressed your concerns, we sincerely hope you would consider updating your rating. If not, please don’t hesitate to let us know your questions, and we are happy to engage in further discussion.

---

> > > > ### Comment · Reviewer_Paj7 · 2025-08-08
> > > >
> > > > Thank you for your reply.
> > > >
> > > > * Regarding this point: "In the rebuttal, we further demonstrate that, for autoregressive models, aligning with pretrained representations (e.g., DINOv2 and MAE) is less effective compared to the end-to-end joint training of ST-AR," I quite agree with it. ST-AR **represents a further step forward compared to directly aligning with DINOv2 and MAE**.
> > > >
> > > > * I have upgraded my rating.

---

### Official Review · Reviewer_rKee · 2025-07-03

**Clarity:** 3
**Significance:** 2
**Originality:** 3
**Rating:** 4
**Confidence:** 2

**Summary:**

This paper presents Self-guided Training for AutoRegressive models (ST-AR), which explores using next-token prediction paradigm to the visual domain. It does not rely on pre-trained representation models (unlike existing approaches such as llava), it enhances image understanding ability, compared to LlamaGen-L and LlamaGen-XL.

**Questions:**

(1) The paper mostly compares with LLamaGen, could the author provide additional baseline?
(2) Could the author discuss how to apply ST-AR to pre-trained representation?
(3) Could the author additionally provide result on more dataset?
(4) On line 245 - 246 the author discusses the contrastive loss is added to the 6-th and 18-th layer, can the author provides rational how these are set?

**Ethical Concerns:**

["NO or VERY MINOR ethics concerns only"]

**Final Justification:**

As the initial review points out, the paper is in good quality and insightful. During rebuttal, the author further addressed my non-major concerns, Thus I would like keep my original rating and vote for acceptance.

**Limitations:**

Yes.

**Paper Formatting Concerns:**

No major problem.

**Quality:**

3

**Strengths And Weaknesses:**

Strengths:
- The paper is well-written and easy to follow.
- The paper is insightful, identifying three properties that impact visual understanding, e.g. local and conditional dependence, inter-step semantic inconsistency, and spatial invariance deficiency, and build accordingly.
- The experiments results are great (Table 1).
- The ablation is comprehensive, covering effectiveness of training losses, mask ratio, contrastive loss depth and number of steps.
- The comparison (on imagenet) is comprehensive and well categorized (Table 2).


Weaknesses:
- The paper should could include comparison to more pre-trained based methods, and more baseline beyond LlamaGen.
- The paper could include more datasets than Imagenet.

---

> ### Author Rebuttal · Authors · 2025-07-31
>
> We truly thank the reviewer for your detailed review and thoughtful comments in helping us enhance the quality of our paper. Below, we provide detailed point-to-point responses addressing each of your valuable suggestions:
>
>
> **Q1: Comparison on other baselines and datasets.**
>
> In the supplementary material, we provide the results of applying ST-AR on the vanilla Transformer, also in Table A1. Besides, we provide the text-to-image generation results of ST-AR and LlamaGen in the supplementary material, also in Table A2.
>
> For the vanilla autoregressive Transformer, we follow the training setup of LlamaGen and conduct comparative experiments on the ImageNet-256x256 benchmark. As shown in Table A1, ST-AR consistently and significantly improves the image generation performance of the vanilla Transformer.
>
> **Table A1: Results of vanilla Transformers on ImageNet-256x256 Benchmark.**
> ||Model|Epochs|FID|sFID|IS|Prec.|Rec.|
> |-|-|-|-|-|-|-|-|
> |w/o CFG|Transformer|50|35.30|6.66|31.90|0.56|**0.59**|
> ||**+ST-AR**|50|**29.37**|**6.08**|**38.88**|**0.60**|**0.59**|
> |w/ CFG|Transformer|50|9.67|6.94|129.50|**0.84**|0.37|
> ||**+ST-AR**|50|**6.86**|**6.40**|**159.13**|0.83|**0.43**|
>
> For text-to-image generation experiments, we train LlamaGen-XL and ST-AR on a 2M subset of the SAM-11M dataset with an image resolution of $256 \times 256$. We use a pretrained FLAN-T5-XL to extract text embeddings, where the maximum embedding length is set to 120.
> We use FID and CLIP Score to evaluate image quality and text-image alignment on the COCO-val-30K benchmark. As shown in Table A2, ST-AR also achieves significant improvements on text-conditional generation, further demonstrating the generalizability of ST-AR.
>
> **Table A2: Results of LlamaGen-XL on the text-conditional COCO-val benchmark.**
> |Model|Epochs|FID|CLIP|
> |-|-|-|-|
> |LlamaGen-XL|50|17.08|0.25|
> |**+ST-AR**|50|**13.52**|**0.29**|
>
> **Q2: Comparison with pretraining-based methods**
>
> **Aligning the intermediate features of autoregressive models with pretrained representations introduces spatial misalignment.** Specifically, for position $i$, pretrained vision encoders extract features of the $i$-th image patch, whereas autoregressive models take the $(i-1)$-th image token as input to predict the $i$-th token. Such behavior is fundamentally distinct from diffusion models.
> Therefore, directly applying representation alignment methods, such as REPA[1], to autoregressive models is **suboptimal**.
> To validate this, we apply REPA to LlamaGen, strictly following the settings of REPA, aligning the intermediate features of LlamaGen-B with representations from DINOv2-B or MAE-B.
> As shown in Table A3, while REPA achieves better generation performance compared to the baseline, it is still inferior to ST-AR.
>
> **Table A3: Comparison between ST-AR and pretraining-based methods.**
> |Model|Pretrained Encoder|Epochs|FID|sFID|IS|Prec.|Rec.|
> |-|-|-|-|-|-|-|-|
> |LlamaGen-B||50|31.35|8.75|39.58|0.57|0.61|
> |LlamaGen-B + ST-AR||50|26.58|7.70|49.91|0.60|**0.62**|
> |LlamaGen-B + REPA|DINOv2-B|50|28.50|7.78|44.72|0.59|**0.62**|
> ||MAE-B|50|31.27|9.12|40.27|0.58|0.60|
> |LlamaGen-B + REPA + ST-AR|DINOv2-B|50|**25.11**|**7.37**|**50.52**|**0.61**|**0.62**|
>
> **Q3: Applying ST-AR to pretrained representations.**
>
> We also apply ST-AR to pretrained representations (the last row of Table A3). We align the intermediate features from the $6$-th layer of LlamaGen-B with the visual representations extracted by pretrained DINOv2.
> We retain the three losses of ST-AR and set the weight of the alignment loss to $0.5$.
> As shown in Table A3, by incorporating additional pretrained representations, the performance of ST-AR is further improved. We attribute this improvement to the richer semantic information learned by DINOv2 on a larger-scale dataset.
>
> **Q4: Choice of contrastive loss depth.**
>
> **The contrastive losses in ST-AR are designed to align the high-level semantic information across different timesteps and views.**
> In Table 5 of our manuscript, we explore the impact of contrastive loss depth and demonstrate that applying the losses at half of the model depth achieves the best performance for ST-AR. Here we provide more insights about this design. REPA [1] and REPA-E [2] propose to divide a diffusion model into an encoder (shallow layers) and a decoder (deeper layers),  where the encoder implicitly learns a representation that reconstructs the target.
> We find that the above perspective also holds true in autoregressive models. Table A4 shows the linear probing performance of features extracted from different layers of LlamaGen-B, where the $6$-th layer achieves the best representation quality.
> This demonstrates that autoregressive models exhibit behavior similar to diffusion models.
> Specifically, for step $i$, the shallow subnetwork (encoder) of the autoregressive model maps preceding tokens into a high-level representation, which is then decoded by the deeper subnetwork (decoder) to predict the $i$-th token.
> Therefore, applying contrastive losses ($L_{step}$ and $L_{view}$) to the intermediate features is an effective strategy.
>
> **Table A4: Linear probing results of different layers in LlamaGen-B.**
> |Layer Index|3|6|9|12|
> |-|-|-|-|-|
> |LP Acc.(%)|8.42|**18.68**|16.90|12.39|
>
> *Reference*
>
> [1] Representation Alignment for Generation: Training Diffusion Transformers Is Easier Than You Think, ICLR 2025
>
> [2] REPA-E: Unlocking VAE for End-to-End Tuning with Latent Diffusion Transformers, ICCV 2025

---

> > ### Comment · Reviewer_rKee · 2025-08-01
> >
> > Thank you for the rebuttal! I will keep the original score and champion for acceptance.

---

> > > ### Author Response · Authors · 2025-08-05
> > >
> > > Thank you for your positive feedback. We hope we have fully addressed your concerns. If you have any further questions, please feel free to let us know.

---

### Decision · Program_Chairs · 2025-09-17

**Decision:**

Accept (poster)

**Comment:**

This paper proposes a framework ST-AR for self-supervised training of autoregressive models in the image domain. The approach is motivated by three issues with existing approaches (local and conditional dependence; inter-step semantic inconsistency; and spatial invariance deficiency), which are addressed in ST-AR with corresponding losses.

Reviewers are quite positive about this work, listing the motivation, writing, effectiveness of the method, and quality of the experiments as strengths. The main limitation of ST-AR is that it increases computational cost, which the authors further investigated in their rebuttal. Another concern is that, although ST-AR is effective, it is a recombination of existing ideas and therefore less novel. That said, after rebuttal, all of the reviewers are in favor of acceptance and I agree that this paper makes a valuable contribution.

I note that one reviewer, reviewer gtuF, did not participate in in the author-reviewer discussion, which was taken into account before making a recommendation.